# Analysis of Selective Pressure on Ancient Human Mitochondrial Genomes Reveals the Presence of Widespread Sequencing Artefacts

**DOI:** 10.3390/ijms26167739

**Published:** 2025-08-11

**Authors:** Pedro Fernandes, Bernardo Pinho, Bárbara Miguéis, João B. Almeida, Teresa Rito, Pedro Soares

**Affiliations:** 1Centre of Molecular and Environmental Biology (CBMA), Department of Biology, University of Minho, Campus de Gualtar, 4710-057 Braga, Portugal; id9568@alunos.uminho.pt (P.F.); bernardopinho@gmail.com (B.P.); a99196@alunos.uminho.pt (B.M.); id10671@alunos.uminho.pt (J.B.A.); teresarito@bio.uminho.pt (T.R.); 2Institute of Science and Innovation for Bio-Sustainability, University of Minho, 4710-057 Braga, Portugal

**Keywords:** ancient DNA, mitochondrial DNA (mtDNA), synonymous mutations, non-synonymous mutations, pathogenicity, quality assurance

## Abstract

Human mitochondrial DNA (mtDNA) is a relevant marker in evolutionary and population genetics, including ancient DNA (aDNA) research, due to inherent characteristics. However, aDNA is prone to damage and sequencing artefacts, potentially confounding evolutionary interpretations. To assess evolutionary patterns in ancient and modern mtDNA, we built a phylogeny comprising 63,965 modern and 3757 ancient public mitogenomes, classified mutations by genomic region and functional effect, and analysed distribution, frequency, and predicted pathogenicity of private and pre-terminal mutations, investigating purifying selection. We compared mutation class ratios (non-synonymous, rRNA, tRNA, nonsense vs. synonymous) across ancient and modern terminal branches and pre-terminal nodes. The predicted pathogenicity of non-synonymous mutations was evaluated across major European haplogroups using three tools. Ancient variants exhibited higher ratios of potentially deleterious mutations and significantly elevated pathogenicity scores compared to modern and pre-terminal branches, highlighting a mutation load likely inflated by damage-related artefacts. Remarkably, nonsense mutations—largely incompatible with life—were over 70 times more frequent in aDNA. The correlation between mutation ratios and predicted deleteriousness across haplogroups suggests a pattern incompatible with biological persistence or relaxed selection alone. These findings highlight the importance of rigorous quality control for ancient data in evolutionary inference, molecular clock calibration, and pathogenic variant identification.

## 1. Introduction

Over the past decade, the use of ancient DNA (aDNA) recovered from fossil remains has increased dramatically, particularly in population genetics and phylogeography. However, aDNA is usually highly fragmented, with limited amounts of endogenous molecules present in ancient remains. The lack of repair machinery capable of countering damage induced by microorganisms and environmental factors through time, such as pH, radiation, temperature, and water, severely affects the integrity of the genome, which accumulates chemical post-mortem damage [1,2]. aDNA sequences are affected by hydrolytic and oxidative reactions that perpetuate modifications such as depurination (the loss of a purine base), contributing to the destabilization of the DNA structure and the formation of shorter fragments with single-stranded overhanging ends. In turn, the ends of fragments become more prone to deamination of cytosine to uracil, which translates as C to T or G to A transitions during amplification [3,4,5]. In addition, the low amounts of endogenous molecules available can make aDNA studies highly susceptible to modern DNA contamination [6,7].

These specific characteristics of aDNA challenge the capacity of researchers to isolate ancient endogenous molecules, which explains the majority of human aDNA research, prior to the advent of Next-Generation Sequencing (NGS), focusing on mitochondrial DNA (mtDNA) [8]. The relevance of mtDNA as a tool for ancient population studies comes from the high copy number of this genetic marker relative to the nuclear DNA, which often facilitates the recovery of mtDNA from archaeological remains [8,9]. Its high mutation rate, maternal inheritance, and lack of recombination allow us to trace back sex-specific demographic events that have impacted the genetic variation in modern day populations [10,11,12]. This approach, of using non-recombining uniparental genetic markers, like the mtDNA, to discern different dimensions of the human past, is termed phylogeography [13,14]. Age estimates of lineages, alongside past dispersals, may be determined by the usage of a mutation accumulation rate (molecular clock), a methodology usually applied to modern DNA data [15,16], but anchored on aDNA variation in some existing approaches [17]. The mutation rate can be estimated using two main strategies. Node-dating is a method for molecular clock calibration that relies on modern DNA data and uses fossil evidence to impose minimum age constraints on internal nodes of a phylogeny [18]. Therefore, this method involves an estimation based on the distance between humans and other species such as chimpanzees, alongside fossil evidence for their divergence time [19]. Such estimates can be further corrected for purifying selection [13], considering the higher frequency of non-synonymous mutations and RNA variants in younger branches of the evolutionary tree, when compared to older ones [20,21]. Most likely, purifying selection acts over time on weakly deleterious mutations that might endure for a certain period within the population. Another commonly used approach to estimate the mutation rate involves the comparison between ancient and modern genomes, without the requirement for an external inferred split time [22]. Tip-dating often yields faster rates and more recent estimates for demographic events in human evolution [17,23]. This method relies on the dating of ancient samples to calibrate the phylogeny by comparing branch length differences with those observed in phylogenetically close modern samples [24,25].

Beyond its application in phylogeography, the non-recombining properties of mtDNA offers a solid basis for testing models of purifying selection as mutations can be hierarchized, allowing to evaluate the higher percentage of non-synonymous mutations near the tips [13] or the presence of variants that are more likely to be pathogenic [26,27].

Our goal is to assess whether observed mutational patterns in ancient mtDNA reflect biological processes such as purifying selection or are instead influenced by post-mortem DNA damage and sequencing-related artefacts inherent to ancient DNA analysis. Given the hierarchization of mutations, private mutations can be present either in modern or ancient mitogenomes. We will analyse ratios of various classes of mutations (non-synonymous and nonsense mutation in coding genes and mutations in tRNA and rRNA genes) against neutral synonymous mutations and use three algorithms of pathogenicity prediction to evaluate how damaging mutations could be in both groups.

Our study revealed a greater presence of potentially harmful mutations with higher pathogenicity in ancient variants, accompanied by significantly increased ratios that exceeded those observed in modern samples. These findings point to a substantial prevalence of sequencing artefacts in ancient samples.

## 2. Results

### 2.1. Higher Ratios of Ancient Terminal Variants: Mutation Load Characterized by “Phantom Mutations”

We assessed the mutational landscape among the different groups of mutations. In the coding region, we identified a total of 3013 terminal mutations in aDNA samples, 21,478 terminal mutations in modern samples, and 16,463 pre-terminal branch mutations. Terminal mutations per sample were more frequent in aDNA data (1.072) than in modern samples (0.593) (*p*-value < 0.0001). This pattern observed in ancient samples may reflect a deeper evolutionary time, resulting in longer terminal branches and a stronger effect of purifying selection. However, it could also be influenced by higher errors, which are ultimately related to damage and artefacts.

Notably, the ratio of non-synonymous or other mutations to synonymous mutations is consistently higher in ancient mtDNA samples when compared to pre-terminal mutations and modern ones (Figure 1). Slightly deleterious mutations may persist in young clades, disturbing the rate between synonymous and other mutations, which may explain the significant difference observed between terminal modern and internal node mutations [13,28]. However, this effect is much more pronounced in terminal ancient mutations, where the ratio is twice as high as in modern terminal and pre-terminal samples for rRNA (Figure 1a), tRNA (Figure 1b), and non-synonymous mutations (Figure 1c). While rRNA, tRNA, and non-synonymous variants can range from basically neutral to highly deleterious, nonsense/stop mutations, which truncate the protein, are inherently harmful and incompatible with survival. The presence of nonsense mutations in both modern and ancient samples is much more likely due to sequencing errors than genuine variation. Although modern data are not error-free (eight stop mutations in 36,227 samples), the proportion is 70 times higher in aDNA (44 stop mutations in 2818 samples), with an even more drastic difference in the ratio against synonymous mutations (Figure 1d). These aDNA mutations passed the detailed curation of the public database, including damage pattern and read depth thresholds. This highlights the significantly higher rate of biologically unreliable variation in aDNA data.

Comparisons between ancient terminal mutations and the modern terminal/pre-terminal groups are always significant (*p*-values < 0.0001), highlighting the distinction of rates in ancient samples. Ratios are always lower for pre-terminal than modern terminal groups (although not significant for rarer tRNAs and nonsense mutations), suggesting that purifying selection has already acted on deleterious characters in pre-terminal. This inferred difference underscores the sensitivity of our methodology.

Ratios from ancient terminal mutations greatly exceed those from modern variants—sometimes more than twofold—undermining the idea that the presence of deleterious characters results from relaxed selection in ancient periods. Instead, these findings strongly suggest a high level of sequencing artefacts generating “phantom mutations” [29], including numerous mutations incompatible with life itself, as seen in the extreme ratios for stop mutations.

We calculated mutation ratios for individual haplogroups (Appendix A), with most results aligning with the main findings. However, haplogroups N1, N2, and X showed non-significant differences in tRNA, rRNA, and non-synonymous to synonymous mutation ratios, despite keeping the overall trend of higher ratios in ancient mutations. This lack of significance may result from smaller sample sizes. Additionally, we compared mutation ratios between haplogroups, with most differences being non-significant (Appendix A). A few significant results emerged, likely due to random effects from multiple comparisons rather than true biological differences. Our analysis confirms that mutation ratios are consistent across and within haplogroups.

### 2.2. Greater Pathogenicity Scores in Ancient Terminal Mutations

We assessed the pathogenicity of non-synonymous mutation using three predictive tools: PolyPhen-2, SIFT, and MutPred2. The distribution of pathogenicity scores of all amino acid variants for haplogroups U, JT, R0, N1, N2, and X is provided in Appendix A. Since purifying selection is expected to remove deleterious mutations over time, older branches should carry fewer damaging variants. However, adding to that, our results show a consistent trend of higher average pathogenicity scores in ancient terminal mutations, regardless of the software used. Modern terminal mutations also display significantly higher scores than pre-terminal variants, as expected (Figure 2).

Polyphen-2 pathogenicity scores have an average of 0.5448 for ancient terminal mutations, while the pre-terminal and modern terminal modifications have averages of 0.2963 and 0.3618, respectively (*p*-values < 0.0001 across the three comparisons). The distribution follows a quasi-bimodal pattern, with most scores clustering near 0 (benign) or 1 (highly damaging). The SIFT results show a similar trend, with ancient terminal mutations averaging 0.2239 (closer to the deleterious threshold of 0.05), while pre-terminal (0.3455) and modern terminal (0.3123) mutations show significantly higher tolerated scores (*p* < 0.0001 for all comparisons). The MutPred2 scores also confirm this pattern, with ancient terminal mutations exhibiting the highest scores (0.3507), followed by modern terminal (0.279) and pre-terminal mutations (0.2531). These results consistently indicate that ancient terminal mutations harbour a greater proportion of potentially deleterious variants, including many with extremely high potential for pathogenicity.

### 2.3. Ancient Terminal Variants Exhibit Consistently Higher Ratios and Pathogenicity Across Haplogroups

Finally, to further investigate the observed trend, we compared the ratio values of non-synonymous to synonymous mutations per haplogroup per group with their average pathogenicity scores (Figure 3). A strong correlation emerges across all haplogroups, with ancient terminal mutations consistently showing the highest ratios and pathogenicity scores. In PolyPhen-2, pathogenicity values exceed 0.50 in all haplogroups (except R0 with ~0.40). MutPred2 shows that even the lowest ancient terminal scores exceed the highest pre-terminal and modern terminal values (e.g., 0.402–0.443 in R0, N2, and X). The SIFT results reinforce this trend, with ancient terminal mutations showing greater predicted pathogenicity, reaching values as low as 0.137–0.145 in haplogroups X, N2, and R0.

In the three analyses presented here, it is also evident that, on the whole, pre-terminal and modern terminal mutations cluster together, while ancient terminal mutations display a more divergent and heterogeneous distribution. This is inconsistent with expectations under purifying selection and suggests stochastic noise. Terminal mutations (modern or ancient) tend to have slightly higher pathogenicity scores than pre-terminal variants, reinforcing the idea that purifying selection is actively shaping mtDNA evolution.

## 3. Discussions

Human mtDNA is a key marker in population genetics due to its maternal inheritance, lack of recombination, and high mutation rate, making it ideal for phylogenetic reconstruction and demographic inference. The reconstruction of reliable and realistic phylogenies, with the hierarchization of mutations, makes mtDNA a powerful model for studying purifying selection, beyond the marker itself. Purifying selection acts gradually over time to eliminate the majority of mutational events in a population, with only a small subset of modifications being retained. These surviving alterations are typically regarded as neutral due to their minimal or non-existent effects on fitness [30]. This effect is clear in the mtDNA phylogeny, both quantitively, with a higher proportion of non-synonymous mutations near the tips of the trees [13,21,26], and qualitatively, as a significant share of these mutations are deleterious [26,27]. Such trend is difficult to evaluate in the recombining autosomal DNA. Nonetheless, the presence of artefacts in autosomes is equally expected considering the inherent degradation of aDNA.

By comparing modern and ancient mtDNA terminal branches, we observed not only longer branches in aDNA, which could be indicative of deeper evolution, but also a significantly higher frequency of non-synonymous, rRNA, tRNA, and stop codon mutations across six European mtDNA haplogroups (R0, U, JT, N1, N2, and X). A first potential alternative explanation is that aDNA harbours more relaxed purifying selection. However, as these ancient mutations belong to the same time range as modern terminal mutations, this is unlikely and many variants are incompatible with survival, by impairment of mitochondrial activity [26,31].

It is more likely that many are artefacts of post-mortem DNA damage rather than true evolutionary changes. This is further supported by the high ratio of non-synonymous to synonymous mutations in aDNA, which reaches values close to 1, mimicking selective neutrality—suggesting randomness rather than biological persistence and the elevated proportion of stop codon mutations, nearly absent in modern samples. Additional evidence strengthening this hypothesis comes from the observation that higher pathogenicity scores are consistently associated with ancient terminal mutations, independent of the prediction software used. A strong correlation between non-synonymous/synonymous ratios and pathogenicity scores further highlights the abnormality of the ancient terminal group, as its results appear dispersed compared to those of the modern terminal and pre-terminal groups.

This issue has broader implications. Pathogenicity prediction tools (e.g., SIFT, MutPred2, PolyPhen-2) often yield false positives, making integrative approaches essential for improving accuracy. By applying multiple tools, we detected consistent trends, strengthening the reliability of our findings. While predicting disease-associated variants is crucial in clinical genetics [32], our focus was on general evolutionary trends, validating the use of phylogenetic reconstruction to assess mutational load. The differences observed between modern terminal and pre-terminal branches also provide general support for the prediction tools—insights that only a non-recombining marker can offer.

In evolutionary analysis, there are implications for mtDNA genetics and beyond. One major application of ancient mtDNA is in tip-dating for molecular clock calibration [33], used to estimate mutation rates without relying on external divergence dates like fossil records. This method assesses variation between dated ancient and modern samples without an inferred split time. However, because terminal branches in mtDNA phylogenies are typically short, even a small proportion of sequencing errors can inflate mutation rate estimates, leading to underestimated divergence times. Notably, the fastest mtDNA mutation rate estimates come from aDNA data [17,23,34]. These estimates can also vary significantly depending on sample size and type, implying some randomness into results. This issue affects both mtDNA and autosomal mutation rate estimates, emphasizing the need for caution when incorporating aDNA into evolutionary time scale studies. Without any doubt, regarding the haploid markers (both mtDNA and the Y chromosome), the hierarchy of the mutations allows lineages to be inferred from a pre-defined phylogeny, even in the presence of artefacts, providing valuable information on the distribution and time of dispersal of lineages in the past.

While mtDNA serves as a model for variant hierarchization, our focus is on broader properties of aDNA sequencing, even while acknowledging biochemical differences between mitochondrial and autosomal DNA. Despite its potential for reconstructing population history, aDNA currently lacks the reliability needed for the precise identification of rare or unique mutations. These artefacts can potentially distort population genetic measures applied to autosomes (e.g., heterozygosity, Watterson’s θ, π, FST), complicate analyses like PCA and selection scans, and cast doubts on estimated frequencies of disease or susceptibility-related variants in past populations.

Although sequencing challenges are expected in aDNA, caution is warranted when interpreting data in public genomic databases. Reliability of individual mutations should be assessed, whenever possible, using software that predicts functional effects or by cross-referencing with modern genomic databases. For instance, in this mtDNA analysis, 991 (32.89%) of the ancient terminal coding region mutations were absent in approximately 70,000 modern samples and over 11,000 different variants. Among these, 118 mutations were in tRNA genes, 221 in rRNA genes, and 647 in protein-coding genes. Within the latter, 44 were nonsense, 445 were non-synonymous, and only 158 were synonymous mutations. The average pathogenicity scores for these novel non-synonymous mutations were notably high: 0.80 (PolyPhen-2), 0.11 (SIFT), and 0.48 (MutPred2). While some variants are putatively variants still unsampled in modern samples or population-specific variants that were subsequently lost through genetic drift, the high pathogenicity probabilities of the unsampled variants strongly supports the high frequency of artefacts. Incorporating robust quality control measures could enhance not only downstream bioinformatics analyses but also improve sequencing accuracy, variant calling, and genome assembly.

## 4. Materials and Methods

We collected a database of all published mitogenomes from NCBI (https://www.ncbi.nlm.nih.gov/), from 1980 until the 19th of January 2024, which comprised 63,965 modern samples, including 2226 sequences from the 1000 Genomes Project [35]. We classified all sequences into haplogroups using Haplogrep 3 v3.2.2 [36], and performed an initial quality control that removed poor quality sequences and individuals with low-confidence haplogroup assignment (less than 80% confidence). For this modern dataset, we also excluded ancient samples, sequences from archaic hominins (Neanderthals and Denisovans), and sequences associated with tumours/cancers or mitochondrial-associated diseases. Furthermore, we included ancient mtDNA sequences for the phylogenetic analysis, obtained from the Allen Ancient DNA Resource (AADR) (https://doi.org/10.7910/DVN/FFIDCW) [37]. This public repository contains published sequences that have passed strict quality control and filtering by the original authors and by the AADR curators, as this is arguably the leading reference database for ancient genomics. Sequences were downloaded as a single file and transformed into a multifasta following instructions on the website. Ancient mitogenomes underwent a similar quality control process as modern DNA. Only samples that were complete and had a haplogroup assignment confidence above 80% were retained. This reduced the dataset to 3757 ancient sequences, ensuring that observed differences against modern mitogenomes were not due to low-quality sequences. All modern and ancient mitogenomes used in this study can be found in Appendix A, respectively.

We performed the phylogenetic reconstruction using an in-house developed software that follows the principles of maximum-parsimony tree construction. The software tool does not reconstruct the full mtDNA phylogeny but uses the inferred haplogroup assignment from Haplogrep 3 v3.2.2 as a guide and uses parsimony to reconstruct the downstream phylogeny. This software was developed to accept as input a template tree acquired from https://github.com/seppinho/haplogrep-cmd (accessed on 11 March 2023), which corresponds to the current version of PhyloTree mtDNA tree, Build 17 [38]. The software also requires a CSV of samples as input containing the information “SampleID, Haplogroup, Not_Found_Polys, Remaining_Polys”, corresponding to the output from Haplogrep 3 v3.2.2. We also inspected the reconstructed phylogeny manually. As a proof of concept, we estimated clade ages from modern data using rho (ρ) and validated them against the previous literature using an in-house software [39]. However, considering that PhyloTree is a “static” tree, which is not updated dynamically based on the input data, there may be some limits to the ability of detecting novel branches or identifying potential misassignments. Nonetheless, our rigorous filtering and quality control aim to mitigate these issues and facilitates the efficient processing of thousands of samples. All analyses were performed in a local server.

After reconstructing the mtDNA tree with modern and ancient sequences, we compiled terminal mutations and applied the same protocol to pre-terminal nodes (one node from terminal branches) (Figure 4). Pre-terminal nodes were chosen to test differentiation from terminal ones, as they have undergone some purifying selection, while newly arisen non-synonymous mutations may still retain most deleterious effects. While deeper mtDNA branches have been shown to have fewer non-synonymous mutations and lower deleterious potential [27], our focus is on early evolutionary changes to demonstrate the discrimination capacity of our methodology on a short time scale of evolution.

We selected mtDNA sequences from only the major European haplogroups (U, JT, R0, N1, N2 and X), resulting in the retention of 36,227 modern and 2818 ancient samples. There are two reasons for the restriction of the analysis to European haplogroups only. One is that the great majority of published aDNA mitogenomes belong to these haplogroups, reflecting a significant under-representation of non-European ancient mtDNAs in public databases and ancient DNA studies (in part reflecting differential environmental conditions favouring DNA preservation). Additionally, the European mtDNA phylogeny displays the highest worldwide resolution.

We classified mutations as being control region, protein-coding gene, tRNA, or rRNA. In addition, we evaluated and registered the amino acid change for each position in protein-coding genes (ND1, ND2, COX1, COX2, ATP8, ATP6, COX3, ND3, ND4L, ND4, ND5, ND6, and CYTB), as well as synonymous, non-synonymous, and nonsense mutations. Given the instability of several control region mutations, we opted to consider only the coding region in the analysis [26]. Our analysis avoids pseudoreplication by considering only evolutionarily independent occurrences of mutations within the reconstructed phylogeny.

To compare the three groups of mutations (terminal ancient, terminal modern, and pre-terminal in both ancient and modern samples), we independently determined the ratio between non-synonymous, tRNA, rRNA, and stop codon mutations and synonymous mutations. This normalization of the rate of mutations against the synonymous mutations is the standard approach in selection and evolutionary analyses, as synonymous mutations are mostly neutral. However, we recognize that, in mtDNA, certain synonymous mutations may experience selective pressure due to codon usage bias or constraints on RNA secondary structure, though these influences are typically limited on the evolutionary scale of the analyses. Therefore, these factors do not undermine the validity of our approach. To test for a significant association between the categorical variables mentioned, we calculated the *p*-value of these comparisons using contingency tables and the online chi-square (χ^2^) test calculator, with a level of significance of 0.01.

We also evaluated the ratios within each individual haplogroup, calculating the *p*-value as described above (Appendix A) to verify if the observed patterns are common across the various major clades. We also calculated *p*-values for the comparison of every ratio between pairs of haplogroups (Appendix A). Considering that chi-square tests can overestimate the statistical significance when working with small sample sizes, and that some of the haplogroups used are more diverse than others, leading to different amounts of registered mutations (sometimes limited), we performed all chi-square statistics with Yates correction.

We used three software tools to predict the impact of protein sequence variants (SIFT v6.2.1, MutPred2, and PolyPhen-2) [40,41,42,43]. These tools are a complementary approach to variant impact prediction, being widely used in studies of genetic disease and evolutionary biology. Notably, MutPred2 was also selected due to its effectiveness in detecting signals of purifying selection in mtDNA, which supports our decision to include it once again [26,27]. We annotated the values of pathogenicity given by each software for non-synonymous mutations, the most common class of mutations found, for the 13 proteins encoded by the mtDNA, across all six haplogroups and the three groups of mutations (ancient and modern terminal, and pre-terminal). These tools assess amino acid changes based on evolutionary conservation, structural impact, and functional annotations. This allowed us to evaluate non-synonymous mutations not only by relative proportion but also by their predicted deleteriousness. Pathogenicity scores in the three tools range from 0 to 1. In Polyphen-2 and MutPred2, the higher scores (1 or near 1) indicate stronger effects on protein structure and function. On the other hand, for SIFT scores, values near 0 are considered as being intolerable [44]. We used an unpaired two-tailed *t*-test with Welch’s correction for the different tools to compare the mean values of two independent groups (ancient terminal vs. modern terminal; ancient terminal vs. pre-terminal; and pre-terminal vs. modern terminal). We performed these analyses with Graphpad Prism version 8.0.1 for Windows (GraphPad Software, La Jolla, CA, USA), using a level of significance of 0.05.

We generated three independent graphs with RStudio 4.4.1, comparing the mean values of pathogenicity and the non-synonymous to synonymous rate per group (ancient terminal, modern terminal, and pre-terminal) for each haplogroup. The R code used to produce these graphs is provided in Appendix A.

## 5. Conclusions

Our findings indicate the prevalence of artefact mutations in aDNA, as suggested by the higher pathogenicity scores and ratios found in these variants. While aDNA can be a unique record of the past, giving insights into various aspects of human history and evolution, this inflated mutation load, likely shaped by damage-related artefacts, might compromise the accuracy of several population genetics analyses. Therefore, genetic data obtained from ancient samples must be carefully scrutinized and validated, incorporating strict quality controls to distinguish genuine evolutionary changes from potential artefacts, which may benefit from rigorous filtering, cross-referencing with modern data, and the usage of different bioinformatics tools for assessing the impact of protein sequence variants.

## Figures and Tables

**Figure 1 ijms-26-07739-f001:**
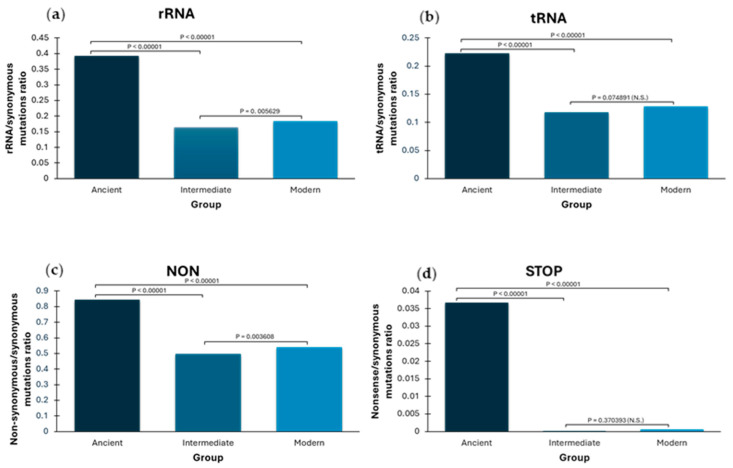
A comparison of the ratios of types of mitochondrial mutations measured against synonymous mutations in ancient and modern terminal samples and pre-terminal branches. Ratios for rRNA (**a**), tRNA (**b**), non-synonymous mutations (**c**), and nonsense mutations (**d**) against synonymous mutations are displayed. *p*-values between ratios in the three analysed groups are shown above the bars.

**Figure 2 ijms-26-07739-f002:**
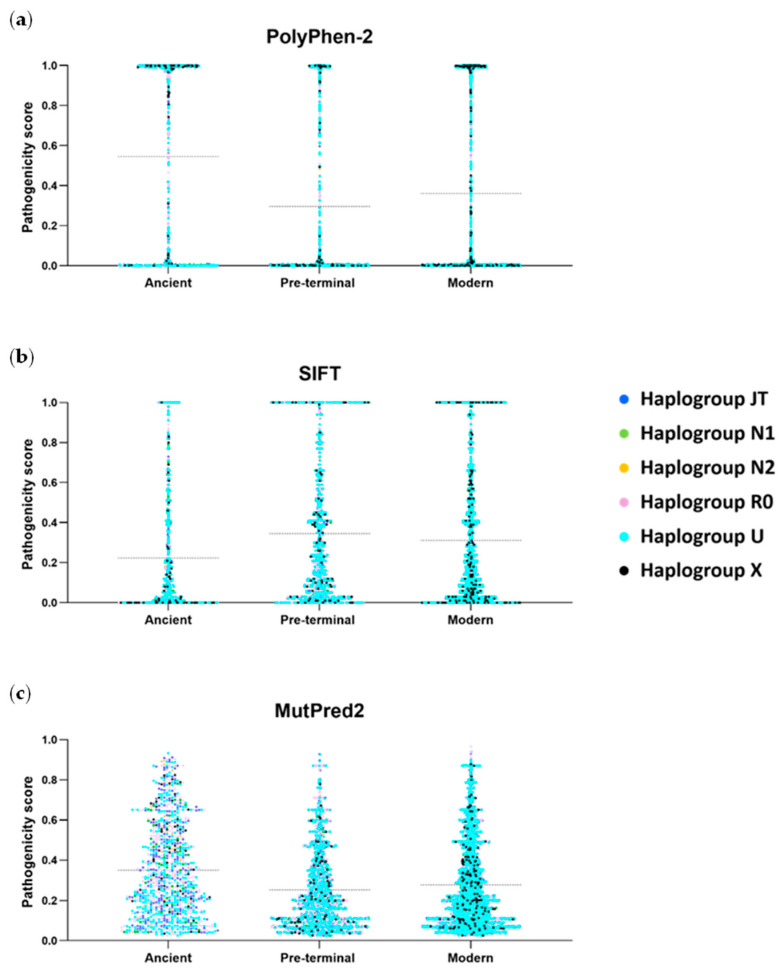
Pathogenicity estimates of amino acid changes across 13 mtDNA-encoded proteins. Pathogenicity scores were obtained using three algorithms ((**a**) PolyPhen-2, (**b**) SIFT, and (**c**) MutPred2) for three classes of mutations defined by the mtDNA human phylogeny: ancient terminal, modern terminal, and pre-terminal branches. Results were obtained from six major European haplogroups. The horizontal dotted line in each distribution of data points indicates the mean values.

**Figure 3 ijms-26-07739-f003:**
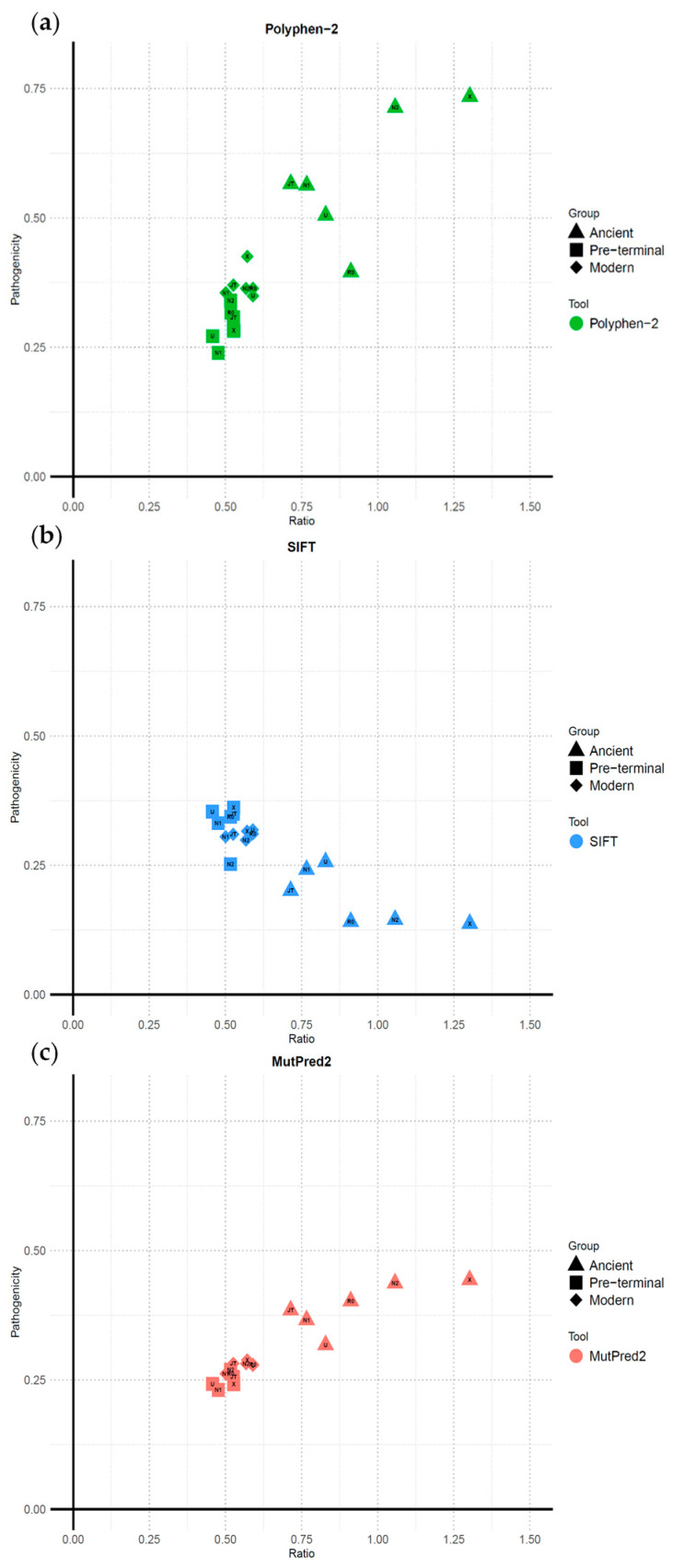
A comparison between the ratio of non-synonymous to synonymous mutations against the average pathogenicity score of non-synonymous mutations using three algorithms, i.e., PolyPhen-2 (**a**), SIFT (**b**), MutPred2 (**c**), for the three classes of mutations (ancient terminal, modern terminal, and pre-terminal) for the six major European mtDNA haplogroups.

**Figure 4 ijms-26-07739-f004:**
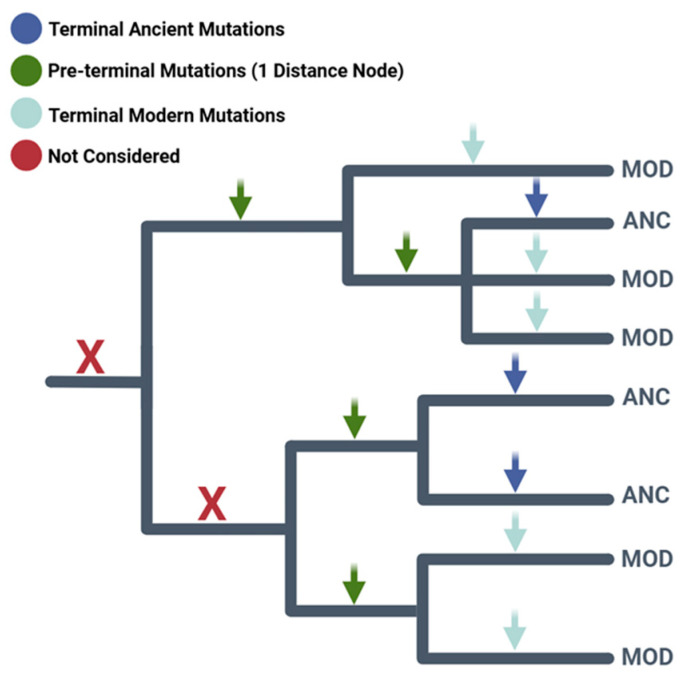
A schematic representation of a phylogenetic tree highlighting the kinds of mutations considered in this study.

## Data Availability

All data needed to evaluate the conclusions in the paper are present in the paper and/or the Appendix A. Any additional information required to reanalyse the data reported in this paper is available from the lead contact upon request.

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
