# Peer review of "Analysis of Selective Pressure on Ancient Human Mitochondrial Genomes Reveals the Presence of Widespread Sequencing Artefacts"

_ijms, 2025, doi:10.3390/ijms26167739_

Round 1
Reviewer 1 Report
Comments and Suggestions for Authors
Manuscript ID: ijms-3768908
Title: Analysis of selective pressure on ancient human mitochondrial genomes reveals the presence of widespread sequencing artefacts
This manuscript presents a timely and important investigation into the mutational landscape of ancient human mitochondrial DNA (mtDNA), with a focus on distinguishing genuine evolutionary signals from potential sequencing artefacts. The authors leverage a large dataset of modern and ancient mitogenomes to assess patterns of purifying selection, mutation load, and predicted pathogenicity. Their central finding—that ancient mtDNA sequences exhibit significantly higher ratios of potentially deleterious mutations, including nonsense mutations incompatible with life—strongly suggests the presence of pervasive damage-related artefacts. This conclusion has significant implications for molecular dating, population genetics, and clinical interpretation of ancient variants. The study is well-conceived, methodologically sound, and addresses a critical issue in ancient DNA (aDNA) research: the reliability of variant calls in degraded samples. The integration of phylogenetic analysis with pathogenicity prediction tools (PolyPhen-2, SIFT, MutPred2) strengthens the robustness of the findings. The manuscript is clearly written, logically structured, and supported by appropriate figures and statistical analyses.
That said, while the overall conclusions are compelling, several aspects of the methodology, interpretation, and presentation could be improved to enhance clarity, rigor, and impact. Below are detailed comments organized by section.
Specific Comments
Abstract
Consider rephrasing “mutation load likely shaped by damage-related artefacts” to “mutation load likely inflated by damage-related artefacts” for precision. “Shaped” may imply a biological role, whereas the authors argue these are non-biological errors.
Introduction
Lines 33–46: The background on aDNA degradation is accurate and well-summarized. However, the description of cytosine deamination could be slightly refined. While it's correct that deamination leads to C→T/G→A transitions, it may be worth noting that this is primarily observed at fragment ends and is asymmetric (more common in single-stranded overhangs).
Lines 58–70: The discussion of molecular clock calibration is relevant but could benefit from a clearer distinction between tip-dating (using ancient DNA directly in phylogenies) and node-dating (using fossil calibrations). This distinction is important for readers unfamiliar with the field.
Line 74: The phrase “Our goal here is to evaluate the effect of selection on modern and ancient DNA samples” is slightly misleading. The study does not directly test selection on ancient populations, but rather evaluates whether observed patterns are consistent with selection or more likely explained by artefacts. Consider rephrasing to: “Our goal is to assess whether observed mutational patterns in ancient mtDNA reflect biological processes such as purifying selection or are instead influenced by technical artefacts.”
Results
2.1. Higher ratios of ancient terminal variants
Figure 1: The visualization is effective. However, the y-axis labels could be improved for clarity. For example, “rRNA/synonymous” might be better phrased as “rRNA : synonymous mutation ratio” to avoid ambiguity.
Line 91: The claim that “terminal mutations per sample were more frequent in aDNA” should be contextualized. This could reflect either longer terminal branches (due to deeper time depth) or higher error rates. The authors correctly interpret this later, but the initial statement may mislead.
Lines 105–108: The observation of 44 nonsense mutations in 2,818 aDNA samples vs. 8 in 36,227 modern samples is striking and well-presented. However, the authors should explicitly state whether these nonsense mutations were filtered out in quality control or whether they passed standard aDNA authenticity criteria (e.g., damage patterns, read depth). This would strengthen the argument that they are likely artefacts despite passing QC.
2.2. Greater Pathogenicity Scores in Ancient Terminal Mutations
Lines 146–157: The use of three independent pathogenicity prediction tools is a major strength. The consistency across tools (PolyPhen-2, SIFT, MutPred2) reinforces the validity of the findings.
The authors should clarify whether pathogenicity scores were averaged per mutation or per sample. Given that some mutations may occur multiple times, it would be important to confirm that the analysis avoids pseudoreplication.
Figure 2: Consider adding individual data points or boxplots to show distribution spread, especially since the text mentions a “quasi-bimodal pattern” in PolyPhen-2 scores.
2.3. Ancient terminal variants exhibit consistently higher ratios and pathogenicity across haplogroups
Lines 167–175: The correlation between mutation ratios and pathogenicity scores is a powerful argument against relaxed selection. However, the interpretation that the distribution is “random” (line 178) may be too strong. A more accurate phrasing might be “inconsistent with expectations under purifying selection” or “suggestive of stochastic noise.”
Supplementary Figures: The supplementary material is extensive and valuable. However, the large number of non-significant comparisons (due to small sample sizes in some haplogroups) could be summarized in a table rather than individual figures to improve readability.
Discussion
Lines 204–208: The argument against relaxed selection is convincing. The authors correctly emphasize that many of the observed variants (especially nonsense mutations) are incompatible with survival. This is a key point and should be highlighted earlier.
Lines 228–236: The implications for molecular clock calibration are critical. The authors rightly caution against uncritical use of aDNA for tip-dating. Consider citing recent studies that have attempted to correct for aDNA damage in mutation rate estimation (e.g., using damage-aware models or uracil-DNA glycosylase [UDG] treatment levels).
Lines 243–249: The extension of findings to autosomal DNA is appropriate but somewhat speculative. While the principles of aDNA damage apply genome-wide, mtDNA has unique properties (high copy number, lack of repair, maternal inheritance) that may amplify artefact visibility. This should be acknowledged.
Lines 253–260: The observation that 32.89% of ancient terminal coding mutations are absent in ~70,000 modern samples is striking. However, some of these may represent rare modern variants not yet sampled. The high pathogenicity scores of these novel variants strongly support the artefact hypothesis, but the authors might consider discussing the possibility of population-specific variants that have been lost due to drift.
Materials and Methods
Lines 282–293: The phylogenetic reconstruction method using Haplogrep and a template tree (PhyloTree Build 17) is practical for large datasets. However, the reliance on a fixed tree topology may limit the ability to detect novel branches or misassignments. The authors should acknowledge this limitation.
Lines 294–301: The definition of “pre-terminal” nodes is clear and biologically meaningful. However, the rationale for focusing only on European haplogroups (lines 302–307) is sound but could be expanded. For example, are non-European ancient mtDNAs underrepresented in public databases? This is likely true but worth stating explicitly.
-Lines 318–324: The use of synonymous mutations as a neutral baseline is standard. However, the authors should note that some synonymous mutations in mtDNA can be under selection due to codon usage bias or RNA structure constraints, though this is likely minor.
Lines 333–349: The use of three pathogenicity prediction tools is commendable. However, the manuscript would benefit from a brief justification of why these specific tools were chosen over others (e.g., CADD, REVEL). Additionally, the authors should clarify whether variants were analyzed in a haplogroup-specific context, as some tools may not account for deep mtDNA phylogeny.
Author Response
Reviewer 1:
Manuscript ID: ijms-3768908
Title: Analysis of selective pressure on ancient human mitochondrial genomes reveals the presence of widespread sequencing artefacts
This manuscript presents a timely and important investigation into the mutational landscape of ancient human mitochondrial DNA (mtDNA), with a focus on distinguishing genuine evolutionary signals from potential sequencing artefacts. The authors leverage a large dataset of modern and ancient mitogenomes to assess patterns of purifying selection, mutation load, and predicted pathogenicity. Their central finding—that ancient mtDNA sequences exhibit significantly higher ratios of potentially deleterious mutations, including nonsense mutations incompatible with life—strongly suggests the presence of pervasive damage-related artefacts. This conclusion has significant implications for molecular dating, population genetics, and clinical interpretation of ancient variants. The study is well-conceived, methodologically sound, and addresses a critical issue in ancient DNA (aDNA) research: the reliability of variant calls in degraded samples. The integration of phylogenetic analysis with pathogenicity prediction tools (PolyPhen-2, SIFT, MutPred2) strengthens the robustness of the findings. The manuscript is clearly written, logically structured, and supported by appropriate figures and statistical analyses.
That said, while the overall conclusions are compelling, several aspects of the methodology, interpretation, and presentation could be improved to enhance clarity, rigor, and impact. Below are detailed comments organized by section.
Specific Comments
Abstract
Consider rephrasing “mutation load likely shaped by damage-related artefacts” to “mutation load likely inflated by damage-related artefacts” for precision. “Shaped” may imply a biological role, whereas the authors argue these are non-biological errors.
We appreciate the reviewer for the positive feedback on our manuscript and for this helpful suggestion. We have revised the sentence in the abstract accordingly. As noted, the term “shaped” could imply a biological role, which is not the intended meaning. The updated phrasing now clarifies that the mutation load is likely inflated by damage-related artefacts, more accurately reflecting the non-biological origin of these observations.
Introduction
Lines 33–46: The background on aDNA degradation is accurate and well-summarized. However, the description of cytosine deamination could be slightly refined. While it's correct that deamination leads to C→T/G→A transitions, it may be worth noting that this is primarily observed at fragment ends and is asymmetric (more common in single-stranded overhangs).
Although we understand the reviewer’s suggestion, we believe we addressed this issue in the submitted manuscript with the sentence “aDNA sequences are affected by hydrolytic and oxidative reactions that perpetuate modifications such as depurination (loss of a purine base), contributing to the destabilization of the DNA structure, and formation of shorter fragments with single-stranded overhanging ends. In turn, the ends of fragments become more prone to deamination of cytosine to uracil, which translates as C to T or G to A transitions during amplification[3–5].”
Lines 58–70: The discussion of molecular clock calibration is relevant but could benefit from a clearer distinction between tip-dating (using ancient DNA directly in phylogenies) and node-dating (using fossil calibrations). This distinction is important for readers unfamiliar with the field.
In response to this suggestion, we have revised the relevant section to provide a clearer distinction between node-dating and tip-dating approaches. We now explicitly describe node-dating as relying on modern DNA and fossil evidence to constrain internal nodes, and tip-dating as incorporating dated ancient samples directly into phylogenies to calibrate terminal nodes.
Line 74: The phrase “Our goal here is to evaluate the effect of selection on modern and ancient DNA samples” is slightly misleading. The study does not directly test selection on ancient populations, but rather evaluates whether observed patterns are consistent with selection or more likely explained by artefacts. Consider rephrasing to: “Our goal is to assess whether observed mutational patterns in ancient mtDNA reflect biological processes such as purifying selection or are instead influenced by technical artefacts.”
We agree with this opinion and have revised the sentence accordingly “Our goal is to assess whether observed mutational patterns in ancient mtDNA re-flect biological processes such as purifying selection or are instead influenced by post-mortem DNA damage and sequencing-related artefacts inherent to ancient DNA analysis.”
Results
2.1. Higher ratios of ancient terminal variants
Figure 1: The visualization is effective. However, the y-axis labels could be improved for clarity. For example, “rRNA/synonymous” might be better phrased as “rRNA : synonymous mutation ratio” to avoid ambiguity.
We appreciate the reviewer’s comment. We changed the y-axis labels accordingly.
Line 91: The claim that “terminal mutations per sample were more frequent in aDNA” should be contextualized. This could reflect either longer terminal branches (due to deeper time depth) or higher error rates. The authors correctly interpret this later, but the initial statement may mislead.
To avoid potential misinterpretation, we have revised the sentence to clarify that the higher average number of terminal mutations in ancient samples may reflect either deeper evolutionary time, resulting in longer terminal branches, or increased error rates related to DNA degradation and sequencing. We added nevertheless that deeper evolutionary times would likely correspond to more intense effect of purifying selection, since this is what we showed that did not occur.
Lines 105–108: The observation of 44 nonsense mutations in 2,818 aDNA samples vs. 8 in 36,227 modern samples is striking and well-presented. However, the authors should explicitly state whether these nonsense mutations were filtered out in quality control or whether they passed standard aDNA authenticity criteria (e.g., damage patterns, read depth). This would strengthen the argument that they are likely artefacts despite passing QC.
As already noted in the original submission (lines 275–280), these ancient mtDNA sequences were obtained from the Allen Ancient DNA Resource (AADR), a curated repository that includes only sequences which have passed stringent quality control by both the original authors and AADR curators. We added: These aDNA mutations passed the detailed curation of the public database, including damage pattern and read depth thresholds.
2.2. Greater Pathogenicity Scores in Ancient Terminal Mutations
Lines 146–157: The use of three independent pathogenicity prediction tools is a major strength. The consistency across tools (PolyPhen-2, SIFT, MutPred2) reinforces the validity of the findings. The authors should clarify whether pathogenicity scores were averaged per mutation or per sample. Given that some mutations may occur multiple times, it would be important to confirm that the analysis avoids pseudoreplication.
Pathogenicity scores were neither averaged per mutation or per sample. To avoid pseudoreplication, pathogenicity scores were calculated for evolutionarily independent mutations only. Mutations were included as separate data points only when they occurred at distinct evolutionary points in the phylogeny, as defined by our reconstruction. Identical mutations inherited within the same clade (i.e., not independent events) were counted only once, regardless of how many samples carried them. We added in the methods: Our analysis avoids pseudoreplication by considering only evolutionarily independent occurrences of mutations within the reconstructed phylogeny.
Figure 2: Consider adding individual data points or boxplots to show distribution spread, especially since the text mentions a “quasi-bimodal pattern” in PolyPhen-2 scores.
Thank you for the comments. We do not fully understand the comment as Figure 2 displays individual datapoints already where the “quasi-bimodal pattern” in PolyPhen-2 scores is clear. Boxplots would not capture that distribution. Violin distribution plots could be more useful but it would not bring much more information in relation to the already displayed individual datapoints. We show the violin distributions plots nevertheless.
2.3. Ancient terminal variants exhibit consistently higher ratios and pathogenicity across haplogroups
Lines 167–175: The correlation between mutation ratios and pathogenicity scores is a powerful argument against relaxed selection. However, the interpretation that the distribution is “random” (line 178) may be too strong. A more accurate phrasing might be “inconsistent with expectations under purifying selection” or “suggestive of stochastic noise.”
We agree that the term “random” may have been too strong in this context. As recommended, we have revised the phrasing to more accurately reflect the intended interpretation, using “inconsistent with expectations under purifying selection” to emphasize that the observed pattern does not conform to a model of strong selective constraint, without implying complete randomness. This change has been implemented in the revised manuscript.
Supplementary Figures: The supplementary material is extensive and valuable. However, the large number of non-significant comparisons (due to small sample sizes in some haplogroups) could be summarized in a table rather than individual figures to improve readability.
We understand the concern regarding readability. However, we believe that presenting these comparisons as individual figures provides a more intuitive and accessible visual overview of the patterns across haplogroups, even when statistical significance is not achieved. Most often, the overall patterns are consistent across haplogroups even if non-significant and that consistency is less clear in a table. Figures allow for immediate comparison between groups, which we feel would be less clear if condensed into a summary table. We have ensured that the figures are clearly labeled and organized to minimize visual clutter and enhance interpretability.
Discussion
Lines 204–208: The argument against relaxed selection is convincing. The authors correctly emphasize that many of the observed variants (especially nonsense mutations) are incompatible with survival. This is a key point and should be highlighted earlier.
We agree that this is a key point, and we had in fact already highlighted it earlier in the manuscript, in lines 102–105 of the submitted draft. Specifically, we stated:
“While rRNA, tRNA and non-synonymous variants can range from basically neutral to highly deleterious, nonsense/stop mutations, which truncate the protein, are inherently harmful and incompatible with survival.”
We believe this early mention helps set the stage for the interpretation presented later in the discussion.
Lines 228–236: The implications for molecular clock calibration are critical. The authors rightly caution against uncritical use of aDNA for tip-dating. Consider citing recent studies that have attempted to correct for aDNA damage in mutation rate estimation (e.g., using damage-aware models or uracil-DNA glycosylase [UDG]
While several well-established tools such as mapDamage/mapDamage2.0, PyDamage, PMDtools, DamageProfiler, are widely used to authenticate, quantify and model ancient DNA damage patterns, others such as Schmutzi or ANGSD are applied to estimate contamination, genotype likelihoods, and improve consensus sequence generation. Together they aim to mitigate post-mortem damage artefacts as much as possible, to improve variant calling accuracy, and reduce the impact of contamination. While we acknowledge that significant efforts have been made over the years to address postmortem damage and contamination, no studies have yet, to the best of our knowledge, applied these damage-aware models explicitly to refine mutation rate estimates in molecular clock calibration.
Lines 243–249: The extension of findings to autosomal DNA is appropriate but somewhat speculative. While the principles of aDNA damage apply genome-wide, mtDNA has unique properties (high copy number, lack of repair, maternal inheritance) that may amplify artefact visibility. This should be acknowledged.
Although we acknowledge the important differences between mtDNA and autosomal DNA, such as mtDNA’s lack of histones, its high copy number per cell, and the fact that its repair mechanisms are less well characterized compared to those of nuclear DNA, it is becoming increasingly evident that mitochondria do possess DNA repair activities, although with notable differences from nuclear repair pathways. However, once an individual has died, all active repair processes cease to exist, and post-mortem degradation affects both nuclear and mitochondrial genomes in broadly similar ways over extended time scales. Notably, the high copy number of mtDNA may even help mitigate the impact of damage-induced artefacts by enabling more reliable consensus sequence reconstruction from multiple template molecules. In contrast, autosomal DNA, with its typically lower copy number, may be even more susceptible to such artefacts during sequencing, potentially amplifying their effects. We therefore believe that our extension of these observations to autosomal DNA, while cautious, is well grounded. We nevertheless added a cautious note: While mtDNA serves as a model for variant hierarchization, our focus is on broader properties of aDNA sequencing, even while acknowledging biochemical dif-ferences between mitochondrial and autosomal DNA.
Lines 253–260: The observation that 32.89% of ancient terminal coding mutations are absent in ~70,000 modern samples is striking. However, some of these may represent rare modern variants not yet sampled. The high pathogenicity scores of these novel variants strongly support the artefact hypothesis, but the authors might consider discussing the possibility of population-specific variants that have been lost due to drift.
Although the percentage of ancient variants found could represent rare modern variants that have not yet been sampled, or population-specific variants that were subsequently lost through genetic drift, the high pathogenicity scores observed combined with the high ratios and the large and globally diverse modern dataset used, indeed supports the artefact hypothesis, suggesting that random drift or limited sampling alone is insufficient to explain their distribution. But it is true that some variants are just unsampled as visible in the fact that some are synonymous mutations. We added extra information in the text.
Materials and Methods
Lines 282–293: The phylogenetic reconstruction method using Haplogrep and a template tree (PhyloTree Build 17) is practical for large datasets. However, the reliance on a fixed tree topology may limit the ability to detect novel branches or misassignments. The authors should acknowledge this limitation.
We have added this clarification to the manuscript to acknowledge this methodological constraint.
Lines 294–301: The definition of “pre-terminal” nodes is clear and biologically meaningful. However, the rationale for focusing only on European haplogroups (lines 302–307) is sound but could be expanded. For example, are non-European ancient mtDNAs underrepresented in public databases? This is likely true but worth stating explicitly.
We appreciate the suggestion to explicitly state the underrepresentation of non-European ancient mtDNAs in public databases.
In our original manuscript (lines 302–307), we have addressed this by explaining that our analysis is restricted to the major European haplogroups (U, JT, R0, N1, N2, and X) primarily because the vast majority of published ancient mtDNA mitogenomes belong to these groups. Additionally, we note that the European mtDNA phylogeny exhibits the highest worldwide resolution, which further justifies this focus.
To clarify and strengthen this point, we now mention the underrepresentation of non-European ancient mtDNAs in public databases and ancient DNA studies in general.
-Lines 318–324: The use of synonymous mutations as a neutral baseline is standard. However, the authors should note that some synonymous mutations in mtDNA can be under selection due to codon usage bias or RNA structure constraints, though this is likely minor.
We agree that while considering synonymous mutations as neutral is a standard approach, some synonymous changes in mtDNA may be subject to selective pressures due to codon usage bias or RNA secondary structure constraints. We have now included a sentence in the manuscript acknowledging this, while emphasizing that these effects are typically small on the analyzed evolutionary scale and do not undermine the validity of our methodology.
Lines 333–349: The use of three pathogenicity prediction tools is commendable. However, the manuscript would benefit from a brief justification of why these specific tools were chosen over others (e.g., CADD, REVEL). Additionally, the authors should clarify whether variants were analyzed in a haplogroup-specific context, as some tools may not account for deep mtDNA phylogeny.
MutPred2, SIFT, and PolyPhen-2 were selected because they offer distinct but complementary approaches to variant impact prediction. Particularly, MutPred2 was previously employed in three related studies by the corresponding author on mtDNA (Soares, P. et al., 2013; Pereira, L. et al., 2011, Pereira et al 2012) where it was effective in capturing purifying selection signals in mtDNA with younger branches displaying greater proportion of potentially deleterious non-synonymous variants and higher ratio of non-synonymous to synonymous mutations when compared to older branches and high pathogenicity scores in oncocytic phenotype in cancer. Therefore, this methodological continuity was practical and scientifically consistent with our previous findings. Polyphen-2 and SIFT are two extensive validated tools widely used in human genetic research (e.g. cancer) to predict deleteriousness. SIFT is based on evolutionary conservation which allows for the detection of deleterious changes assuming that substitutions at highly conserved positions are more likely to be deleterious. MutPred2 models the structural and functional changes which gives insights into the potential mechanisms affected by different mutations. We mostly aimed to provide comparative values, and while CADD and REVEL were possibilities we already offer three comparative tools (which the review called commendable). Regarding the second part of the suggestion, mtDNA-specific phylogenetic depth was not a major concern, as all European haplogroups fall within haplogroup N and do not differ substantially in terms of amino acid changes. However, expanding the analysis beyond Europe—particularly to Africa, where mitochondrial diversity is much deeper, or to regions where divergent clades such as M and early N lineages are found—would require clade-specific analysis.

Reviewer 2 Report
Comments and Suggestions for Authors
This paper examines a huge number of modern and ancient mitogenomes. They noticed that nonsense mutations were 70x more common in aDNA which is a bit odd. This highlights necessary quality control. This is an incredibly important finding for the aDNA community and this paper is incredibly well written and organized.
Introduction. Good description of technical aspects and nice organization. Terms are clearly defined.
Results are well structured and are nicely written.
Material methods are clear as is data sharing. (one small issue I mention at the end here).
Please document r code/packages for creating figures.
One other detail would be how these tests were run, were they done locally on a large RAM computer or was this done via the cloud? I missed this but would be beneficial to readers.
Figure 4 is very nice.
Conclusions. Short but to the point.
Supp data: figures here are nicely laid out, very clearly labeled and readable. Supp table is enormous, hopefully published as an excel file?
The only issue I have, even though it would be a lot to ask, is to have in the supplemental details on all the 65k~ samples used in this study. It would be interesting to highlight those samples that very obviously had more of these nonsense mutations to aid future researchers.
Author Response
Reviewer 2
This paper examines a huge number of modern and ancient mitogenomes. They noticed that nonsense mutations were 70x more common in aDNA which is a bit odd. This highlights necessary quality control. This is an incredibly important finding for the aDNA community, and this paper is incredibly well written and organized.
Introduction. Good description of technical aspects and nice organization. Terms are clearly defined.
Results are well structured and are nicely written.
Material methods are clear as is data sharing. (one small issue I mention at the end here).
Please document r code/packages for creating figures.
We thank the reviewer for the supportive comments on our manuscript and for this suggestion. The R code used to generate all figures is now provided in Supplementary Code Files S1–S4, to ensure reproducibility.
One other detail would be how these tests were run, were they done locally on a large RAM computer or was this done via the cloud? I missed this but would be beneficial to readers.
The tests were conducted locally on a server with a large RAM. We added the information to the text.
Figure 4 is very nice.
Conclusions. Short but to the point.
Supp data: figures here are nicely laid out, very clearly labeled and readable. Supp table is enormous, hopefully published as an excel file?
We thank the reviewer for the positive feedback regarding the clarity and layout of the supplementary figures. We understand the concern about the size of the supplementary table. While we considered submitting it as an Excel file, we felt that including all supplementary material in a single, well-organized PDF would facilitate readability and maintain coherence.
The only issue I have, even though it would be a lot to ask, is to have in the supplemental details on all the 65k~ samples used in this study. It would be interesting to highlight those samples that very obviously had more of these nonsense mutations to aid future researchers.
We appreciate the reviewer’s interest in access to detailed information for all samples. We have included the relevant details for both modern and ancient mitogenomes in Supplementary Tables S2 and S3, respectively.